# Extracellular Vesicles in the Central Nervous System: A Novel Mechanism of Neuronal Cell Communication

**DOI:** 10.3390/ijms25031629

**Published:** 2024-01-28

**Authors:** Francesca Martina Filannino, Maria Antonietta Panaro, Tarek Benameur, Ilaria Pizzolorusso, Chiara Porro

**Affiliations:** 1Department of Clinical and Experimental Medicine, University of Foggia, 71121 Foggia, Italy; francesca.filannino@unifg.it; 2Department of Biosciences, Biotechnologies and Environment, University of Bari, 70125 Bari, Italy; mariaantonietta.panaro@uniba.it; 3Department of Biomedical Sciences, College of Medicine, King Faisal University, Al-Ahsa 31982, Saudi Arabia; tbenameur@kfu.edu.sa; 4Child and Adolescent Neuropsychiatry Unit, Department of Mental Health, ASL Foggia, 71121 Foggia, Italy; ilaria.pizzolorusso3@gmail.com

**Keywords:** extracellular vesicle, microglia, astrocytes, oligodendrocytes, communication, central nervous system cells

## Abstract

Cell-to-cell communication is essential for the appropriate development and maintenance of homeostatic conditions in the central nervous system. Extracellular vesicles have recently come to the forefront of neuroscience as novel vehicles for the transfer of complex signals between neuronal cells. Extracellular vesicles are membrane-bound carriers packed with proteins, metabolites, and nucleic acids (including DNA, mRNA, and microRNAs) that contain the elements present in the cell they originate from. Since their discovery, extracellular vesicles have been studied extensively and have opened up new understanding of cell–cell communication; they may cross the blood–brain barrier in a bidirectional way from the bloodstream to the brain parenchyma and vice versa, and play a key role in brain–periphery communication in physiology as well as pathology. Neurons and glial cells in the central nervous system release extracellular vesicles to the interstitial fluid of the brain and spinal cord parenchyma. Extracellular vesicles contain proteins, nucleic acids, lipids, carbohydrates, and primary and secondary metabolites. that can be taken up by and modulate the behaviour of neighbouring recipient cells. The functions of extracellular vesicles have been extensively studied in the context of neurodegenerative diseases. The purpose of this review is to analyse the role extracellular vesicles extracellular vesicles in central nervous system cell communication, with particular emphasis on the contribution of extracellular vesicles from different central nervous system cell types in maintaining or altering central nervous system homeostasis.

## 1. Introduction

Communication in the central nervous system (CNS) is fundamental for different biological functions including brain development, homeostasis preservation, and neural circuit formation. Indeed, the crosstalk between glia and neurons is critical in the CNS for a variety of biological functions, such as brain development, neural circuit maturation, and homeostasis maintenance. Glia cells are involved in different processes including inflammatory responses to infections or diseases, neurotrophic support, and synaptic remodelling and pruning. In addition to the traditional direct cell-to-cell contact, glial cell can also communicate with neurons through the paracrine action of secreted molecules, or by the release and reception of extracellular vesicles (EVs) [1,2,3]. EVs, which are subdivided into three subtypes: microvesicles, exosomes, and apoptotic bodies, are a major constituent of the cell secretome. EVs have the ability to circulate in the extracellular body fluid and modulate several biological processes and their associated pathways [4,5,6].

EVs cross the blood–brain barrier (BBB) bidirectionally from the bloodstream to the brain parenchyma and vice versa. They play an important role in brain–periphery communication in physiology and pathophysiology. According to the current literature, although EVs cross the BBB, it is unclear how, where, and when they can overcome this tightly controlled cellular barrier [7].

Most in vitro models that have studied this process utilizing a monolayer transwell or microfluidic organ-on-a-chip techniques do not reproduce the combined effects of all cellular layers that participate in the constitution of the BBB at different sites in the CNS [8].

Matsumoto and coauthors have reported that EVs were internalized by all BBB cell types but did not cross the endothelium [9]. Other authors have suggested that EVs could cross the BBB in an inflammatory environment [10,11]. However, when evaluating their uptake and transcytosis through the BBB, research on EVs originating from pericytes, astrocytes, or neurons is lacking. Transcytosis represents one of the primary mechanisms allowing EVs to pass through the healthy BBB.

A recent study found that EVs can interact with CNS barriers in three ways: (1) Barrier cells can release EVs, which can enter the brain parenchyma or travel through the bloodstream carrying their cargo to more distant sites. (2) Brain-derived or circulating EVs may directly influence BBB function and properties. (3) EVs can cross the barrier non-selectively when the BBB is damaged [12].

Peripheral inflammation can also overcome the BBB and reach the CNS via exosomes. In fact, following the intraperitoneal injection of lipopolysaccharide (LPS), serum exosomes enriched in miR-15a, miR-15b, miR-21, and miR-155, cross the BBB and induce microgliosis, with elevated levels of miR-155 and pro-inflammatory cytokines (TNF-α and IL-6) [13]. This confirm that EVs can traverse the BBB in bidirectional ways.

In the CNS, EVs contribute to the maintenance of homeostasis, because they actively participate in waste clearance, trophic support of neurons, antigen presentation, and the maintenance of myelin and synaptic plasticity [14]. Moreover, EVs have been implicated in the pathogenesis of CNS disorders [15] as well as in the communication between the cells in the microenvironment of CNS tumours, such as glioblastomas [16]. It is challenging to assess the impact of EVs on CNS disorders since, on the one hand, EVs may remove toxic proteins and aggregates, while, on the other hand, they may spread pathogenic proteins [17].

Even though not all facets of this novel biological tool have been investigated, the literature on EVs has gained an increased amount of attention in the last two decades. In this review, we have summarized the recent research on the role of EVs in the CNS, with a particular emphasis on the contribution of EVs from different types of CNS cells to the maintenance or disruption of CNS homeostasis.

## 2. Extracellular Vesicles

Both prokaryotic and eukaryotic cell types release EVs in response to specific external signals; the level and the composition of EVs are in a dynamic state, reflecting the parental cells.

Based on their diameter, source, and biochemical markers, three subgroups of EVs are identified: exosomes (Exos), microparticles (MPs)/microvesicles (MVs), and apoptotic bodies.

MPs/MVs (100–1000 nm) are generated by budding of the plasma membrane and Exos (100–200 nm) are derived from the endosomal/multivesicular body (MVB) system and accumulate inside the cell prior to their release; in contrast, cells release apoptotic bodies (1000–5000 nm) during the final phase of the apoptotic process [18].

With respect to the biogenesis there are some differences between exosomes and microvesicles; exosomes are produced by the inward budding of the endosomal membrane, and in this way, intraluminal vesicles (ILVs) are formed. During the maturation they become multivesicular bodies (MVBs), and they are then released upon fusion of the MVBs with the plasma membrane [18]. Different mechanisms and players are implicated in the biogenesis of exosomes such us Rab GTPases, subunits of endosomal sorting complex required for transport (ESCRT), syntetin-1, *tumour susceptibility gene 101* (TSG101), *apoptosis-linked gene 2-interacting protein X* (ALIX), ceramide, sphingomyelinases, and tetraspanins including cluster of differentiation CD 9, CD63, and CD81, which also regulate cargo sorting into the vesicles [19,20,21,22]. Interestingly, it has been demonstrated that the production of exosomes in all brain cell types, at least in part, depends on an ESCRT-independent mechanism [23].

Microvescicles originate from the outward budding of the plasma membrane and, similarly to exosomes, the formation of microvesicles partially depends on ESCRT proteins and the generation of ceramide by sphingomyelinase. Moreover, microvesicles’ biogenesis requires reorganization within the plasma membranes of protein and lipid components, that include flipping phosphatidylserine from the inner leaflet to the cell surface with consequent physical bending in the membrane and the reorganization of the actin cytoskeleton, culminating in membrane budding and vesicle release [18,24]

Due to some overlap in dimensions and markers between MVs and Exos, the International Society for Extracellular Vesicles set up the Minimal Information for Studies of Extracellular Vesicles (MISEV) [25] in 2014 and updated it in 2018 [26].

The MISEV gives information on nomenclature, collection, pre-processing, separation, concentration, characterization, functional studies, and reporting.

Moreover, the Transparent Reporting And Centralizing Knowledge in Extracellular Vesicle Research (EV-TRACK) Consortium suggested a tool named EV-METRIC to ameliorate experimental rigor by using a quality chart with nine parameters to increase transparency in the methods used for EV separation and characterization [27]. These nanoscale vesicles contain proteins, nucleic acids, lipids, microRNAs, and other biological signals generated from parental cells. These bioactive molecules can be taken up and delivered to receptor cells, altering their biological function. Specifically, they transfer both mRNA and miRNA to recipient cells, transforming EVs into a potentially powerful tool for horizontal gene regulation [28,29,30]. Several techniques and technologies, such as differential ultracentrifugation, immune capture, ultrafiltration, and size exclusion chromatography, have been used to identify and separate EVs from biological fluids [31].

These techniques and technologies are based on the physico-chemical properties of the EVs. Atomic force microscopy, procoagulant assays, flow cytometry, and ELISA-based solid-phase capture assays are the key methods used to discover the essential information about the structure of EVs [32,33]. Other imaging techniques, such as the scanning electron microscope (SEM) and the transmission electron microscope (TEM), are used in addition to electron microscopy to analyse all EVs [34]. Many studies have found that EVs in the CNS serve as promising vectors in the pathogenesis of neurological diseases, or intercellular communication, or are used for vaccine and drug delivery. Furthermore, circulating EVs can also be used as biomarkers for the diagnosis and therapeutic follow-up of CNS-associated diseases [35]. EVs can interact with recipient cells in different ways. They can penetrate with different mechanisms, including clathrin-dependent fusion [36,37], macropinocytosis [38], and lipid-raft-mediated endocytosis [39]. EVs can also activate the target cell’s membrane protein [40]; refer to Figure 1.

In clathrin-mediated endocytosis, EVs bind to the plasma membrane and are internalized; inside the cells, they can be degraded by lisosome or escape to lisosome and release their cargo to cytoplasm [36,37]. Macropinocytosis is the most utilized EVs internalization pathway; in this mechanism, the plasma membrane ruffles out in an actin-dependent manner and internalize EVs [38]. Another mechanism of internalization is represented by lipid-raft-mediated endocytosis in which the disruption of these structures by altering cholesterol dynamics induces Ev uptake [39]. EVs may also cause a change in recipient cells without internalization by a bind of receptor present on the plasma membrane and proteins presents on EVs [40].

When EVs are in contact with the target cells, they may influence their biological behaviour in different ways. As signal complexes, EVs have the ability to directly activate target cells. They can also transfer mRNA or assign intracellular proteins to target cells. EVs can also supply receptors and/or exchange bioactive lipids across cells [41]. EVs are released by all cell types in the CNS and can communicate with neighbouring cells or may be released into the cerebrospinal fluid (CSF) and blood. Cells can release EVs in both physiological conditions and in response to precise stimuli. In addition, EVs may deliver their cargo molecules involved in both physiological and pathological processes, from parental cells to the target cells, generating biological responses [42,43].

## 3. EVs in the CNS

Brain-derived vesicles are a heterogeneous population of CNS-derived EVs which are secreted into the extracellular space and can be retrieved in bio fluids [44], including blood [45] and CSF [46]. It is known that EVs mediate the interactions of nervous system cells among themselves, as well as the communication of peripheral organs with the CNS [47,48] Neurons, astrocytes, oligodendrocytes, and microglial cells release EVs and exchange signal molecules through them [49].

EVs have an important role in promoting neural development, regulating the progression of inflammation, and altering tumour characteristics [50,51,52]. Alterations in EVs’ signalling, especially changes in miRNA expression, or the movement of inflammatory molecules may induce variations in the physiological microenvironment, which are closely associated with the degree of central nervous system (CNS) injury [53]. EVs are able to pass the BBB to enter the brain from the periphery or to exit via the circulating [54] CSF or the lymphatic system; due to this, they may be studied as potential biomarkers or therapeutic carriers of CNS diseases [55].

Different studies have highlighted the great potential of the EVs in CNS diseases to play a dual role: on the one hand, cells use EVs to remove toxic proteins and aggregates from their cytoplasm; on the other, these EVs can interact with healthy cells, delivering their toxic cargoes and spreading disease [56]. In the CNS, EVs have been suggested to be potential carriers in the intercellular delivery of misfolded proteins associated with neurodegenerative disorders, such as tau and amyloid β (Ab) in AD, asynuclein in PD, SOD1 in amyotrophic lateral sclerosis, and huntingtin in Huntington’s disease [57,58,59,60,61]. Thus, these recent findings have opened up avenues for the exploitation of these EVs, which may serve as potential biomarkers for different chronic neurodegenerative diseases [62].

Extracting EVs from peripheral blood for the early diagnosis, progression monitoring, and prognosis assessment of CNS diseases may be a safer and more sensitive non-invasive method compared to a “cerebrospinal fluid biopsy” [63,64,65]. Therefore, EVs, serving as biomarkers, reflect changes in the CNS microenvironment, offering new insights for the diagnosis and treatment of CNS diseases [66].

EVs represent the fingerprints of their cells of origin, they can contain and transfer complex molecular cargoes typical of their cells of origin, such as proteins, lipids, carbohydrates, and metabolites, to recipient cells. EVs are also enriched in non-coding RNAs (e.g., microRNAs, lncRNAs, and circRNA), which are formerly considered as cell-intrinsic regulators of CNS functions and pathologies, thus representing a new layer of regulation in the cell-to-cell communication [67].

MicroRNAs (miRNAs) are small non-coding RNAs with a unique ability to control the transcriptomic profile by binding to complementary regulatory RNA sequences. The ability of miRNAs to increase (proinflammatory miRNAs) or reduce (anti-inflammatory miRNAs) inflammatory signalling within the central nervous system is an area of ongoing research, particularly in the context of disorders that feature neuroinflammation, including neurodegenerative diseases [68].

EVs are highly enriched in miRNAs [68]. Recent findings have provided further evidence for the emerging roles of miRNAs in neural development, homeostasis, neuron-glia communications, CNS health, and a range of physiological functions [67]. Table 1 summarizes some of the most important miRNAs contained in EVs and their various functions in brain cells.

Long Non-Coding RNAs (LncRNAs) are also contained in EVs. LncRNAs are molecules that play a key role in the regulation of a wide range of cellular processes, acting as chromatin regulators and regulating gene expression at the transcriptional and post-transcriptional levels [77]; they are a highly heterogeneous class of RNA molecules of more than 200 nucleotides in length with no protein-coding capacity. They are involved in the control of gene expression at multiple levels, such as nuclear architecture, transcription regulation, mRNA splicing, and mRNA stability. Increasing evidence has revealed that lncRNAs can act as ceRNAs via competitively sponging miRNAs to regulate neuroinflammation in several neurodegenerative diseases [78].

Canseco-Rodriguez et al. [79] have described the LncRNA in Alzheimer diseases, and Wang and colleagues have investigated their involvement in Parkinson’s disease [80].

Liao et al. have discovered that lincRNA-Cox2-siRNA-loaded EVs also decreased LPS-induced microglial proliferation in mice. These findings indicate that the intranasal delivery of EV-loaded small RNA could be developed as a therapeutic for the treatment of a multitude of CNS disorders [81].

In addition to miRNAs and lncRNAs, EVs also contain circRNAs that act like ceRNAs to regulate miRNA function in recipient cells. CeRNAs are contained within EVs, and their specific sorting might have two effects: (i) to discard competitors, which modifies the bioavailability of the related active miRNAs, and (ii) to transfer the competitors into recipient cells in which the amount of active miRNA should be altered [82]. The role of ceRNAs carried by EVs has been intensely studied in the field of cancer research; only a few EV-ceRNAs have been reported in neurodegenerative diseases. A recent study revealed that exosomes contain more circRNAs compared to parental cells and the ratio of circRNA level to linear RNA level in exosomes was approximately six-fold higher than that in cells [60,83,84].

EVs can serve as natural carriers for therapeutic agents and drugs, and have many advantages over conventional nanocarriers, including their low immunogenicity, good biocompatibility, natural blood–brain barrier penetration, and capacity for gene delivery [85,86,87].

Wu and colleagues have studied the capacity of engineered EVs encapsulated with Bryostatin-1, a natural compound with remarkable anti-inflammation ability, to reduce neuroinflamamtion [88]. Moreover, engineered EVs charged with nerve growth factor and curcumin, significantly improved the microenvironment after injury and promoted the recovery of motor function after spinal cord injury [89].

The applications of EVs have received much attention and different clinical trials have been registered in recent years; Table 2 reports the most significant clinical trials for EVs reported in https://clinicaltrials.gov (accessed 20 January 2024) [90].

## 4. Neuron-Derived EVs

EVs are multifunctional and complex signalling units that can simultaneously supply recipient cells with a variety of possible effectors [2]. The production of neuronal EVs suggests that they are derived from the endosomal multivesicular body (MVB)-dependent exosome pathway, which uses multiple (possibly overlapping) mechanisms to bud vesicles into the endosomal lumen. Selective cargo loading of neuronal EVs by cone-shaped lipids such as ceramide can occur in both an ESCRT-dependent and ESCRT-independent manner. The best-studied mechanisms are based on ESCRT proteins. The endosomal sorting complex required for the transport (ESCRT) pathway comprises five distinct protein complexes, referred to numerically as ESCRT-0, -I, -II, -III, and the AAA ATPase Vps4 complex. Of these, the ESCRT-0, ESCRT-I, and ESCRT-II protein complexes bundle ubiquitinated cargoes as they curl around membranes, then recruit ESCRT-III components to form a helical polymer that drives the cleavage of the intraluminal vesicle bud in the MVB. While ESCRT’s molecular activities are interesting in explaining their formation, ESCRT components may not always be necessary for EV release. The enzyme sphingomyelinase plays an important role in EV release and can act simultaneously with or independently of ESCRT. There are two types of sphingomyelinases. Neutral sphingomyelinase (n-SMase) cleaves sphingomyelin to release ceramide, which clusters into membrane microdomains that promote inward budding. Acid sphingomyelinase (a-SMase) plays an important housekeeping role in sphingolipid metabolism and membrane turnover. It has also been implicated in the cellular stress response. It can be preferentially transported to the outer leaflet of the cell membrane under conditions of cellular stress. Activated factor VII (FVIIa) induces the release of EVs from the endothelium. FVIIa-released EVs become enriched with phosphatidylserine (PS) and contribute to the haemostatic effect of FVIIa in thrombocytopenia and haemophilia. In the brains of Alzheimer’s patients, for example, the expression levels of a-SMase are increased and the enzymatic activity is abnormally high [98,99,100,101]. It is unclear how cargo is selected during EV biogenesis. It is regulated by post-translational modifications like ubiquitination and sumoylation and seems to rely on interaction with microdomains that are rich in cholesterol and tetraspanin [2]. Given the morphology and compartmentalisation of neurons, understanding the functions of EVs requires knowing from which part of the neuron the EVs are released. Several studies show that EV cargoes and/or MVBs accumulate in the somatodendritic compartment [98]. There is extensive evidence for the neuronal accumulation of EVs in the somatodendritic compartments of mammalian cortical and hippocampal neurons. The superfamily of proteins called tetraspanins, which organises microdomains in the plasma membrane, has a high concentration of exosomes. The tetraspanins CD9, CD63, CD81, and CD151 are involved in exosome biogenesis, protein loading, and sorting cargo into exosomes [102,103]. Furthermore, exosomes are secreted by mature neurons at the soma and dendritic shafts levels, a process that may be part of the physiology of differentiated neurons rather than growth cone development [104]. MVB accumulates in the dendrites of pyramidal, Purkinje, and cortical neurons, suggesting they are released from these sites. Notably, in mammals, a direct mechanism for their preferential release from dendrites is suggested by the movement and local translation of Arc mRNA into dendrites in response to activity-dependent cues [105]. Importantly, EVs’ secretion by neurons may be regulated process that is induced by high K^+^-induced depolarisation, Gamma-aminobutyric acid type A receptor (GABAAR) antagonists, and/or blocked by NMDA receptor blockers [105,106]. There is also evidence that an enhancement in EVs’ secretion is due to an increase in intracellular calcium [103,106]. The Wnt morphogens family orchestrate a myriad of developmental processes, including the control of cell proliferation and migration and cytoskeletal remodelling. Wnts also coordinate key aspects of the nervous system, regulating neural stem cell proliferation, axon pathfinding, synapse differentiation and plasticity, and learning [107]. Wnts activate a variety of intracellular signalling pathways, the most studied of which involves Wnt ligands binding to the Frizzled (Fz) family of serpentine receptors. The activation of Fz receptors in turn stabilises cytoplasmic β-catenin, which enters the nucleus and regulates gene expression. Other Wnt pathways that have been studied include those involving GSK3-β, which acts through a non-genomic mechanism by phosphorylating microtubule-associated proteins to regulate microtubule stability, as well as mechanisms activated by Wnt ligands such as the planar cell polarity (PCP) pathway and the Wnt/Ca++ pathway [40]. Wnts are one of the first molecules identified as being secreted by EVs and to have a signalling function during development, and are among the molecules expressed and incorporated into neuronal EVs. EVs that are taken up influence a wide range of cellular processes, including neuronal activation, axon guidance, synapse formation, maintenance of dendritic spines, synapse elimination, brain development, and cell expression [107]. Over the years, several molecules contained in neuronal EVs have been studied, including recent studies indicating that Wnt-z signalling is also important for synaptogenesis, synapse and dendrite maintenance, spatial learning and memory, and the formation of hippocampal long-term potentiation (LTP) [108,109]. Other studies have shown that the secretion of PRR7 by neurons on exosomes is activity dependent. Proline-rich 7 (PRR7) is a proline-rich type 1 transmembrane protein first identified as a protein enriched in the postsynaptic density, and these findings highlight the signalling function of neuronal exosomes in synapse maintenance in central neurons [110]. PRR7 functions as a novel Wnt inhibitor in synapse regulation by inhibiting Wnt secretion by exosomes. One of the molecules consistently found in neuronal exosome preparations are α-amino-3-hydroxy-5-methyl-4-isoxazolepropionic acid (AMPA) receptors. AMPA receptors control synaptic transmission and are the main substrates for synaptic plasticity. Glutamatergic neurons form AMPA receptor-containing exosomes following bursts of synaptic activity or ionomycin-induced increases in cytosolic calcium levels. They may contribute to neuronal excitability in recipient neurons. However, the role of exosomal AMPA receptors is currently completely unknown [111].

In addition, EVs transport a variety of molecules that regulate synaptic plasticity. These molecules include endocannabinoids, which affect synaptic functions in both short- and long-term forms of plasticity [112]; the Eph receptor tyrosine kinase and its membrane-bound ephrin ligands, which play a critical role in axon guidance and specific synapse formation [113]; and the neurotrophin receptor p75, which mediates multiple signalling pathways for neurite outgrowth, neuronal survival, and death [114].

miRNAs are one of the most studied categories of molecules transported by EVs. These molecules are non-coding single-stranded RNAs with 19–24 nucleotides directly which are involved in post-transcriptional gene silencing [115]. Several reports indicate that neurons may secrete different types of miRNAs, including miR-124, miR-21-5p, and miR132, which are subsequently taken up by microglia, astrocytes, or endothelial cells [116,117]. These exosome-derived miRNAs influence a variety of processes in the recipient cells, including the modulation of microglial activity, pro-inflammatory responses, gene transcription in astrocytes, and brain vascular integrity. Antoniou et al. showed that the brain-derived neurotrophic factor (BDNF) promotes the sorting of miRNAs into neuronal exosomes, which in turn stimulates the formation of excitatory synapses in recipient neurons [118]. Much less is known about the mechanisms by which neurons take up neuronal exosomes.

According to a number of studies, synaptic clefts appear to be too small for EVs to freely enter and diffuse. For this reason, neuronal exosomes are most likely to be taken up at non-synaptic sites, including the extra-synaptic membrane area in dendrites, axons, and soma [106]. Under normal conditions, neuronal EVs are taken up by neighbouring neurons in a juxtracrine and autocrine manner. Certain conditions, such as extracellular space expansion and/or excessive secretion, cause small neuronal EVs to diffuse further away before being taken up by distant neurons. However, ageing causes shrinkage of the ECS, which may restrict the movement of small EVs and interfere with exosome-mediated intercellular communication.

Characterising EVs from the CNS is challenging because neurons generally release low levels of EVs. However, several studies have identified proteomic signatures of EVs that can help identify their most likely origin as being the CNS. Hornung et al. investigated putative neuron-derived exosomes obtained by immunoprecipitation using antibodies directed against the neuronal marker proteins such as NCAM, that is a neuronal cell adhesion protein that belongs to the immunoglobulin superfamily and is involved in cell–cell and cell–matrix interactions. Another neuronal marker protein that Hornung uses is L1CAM, which is an axonal glycoprotein that is essential for the development of the nervous system [119]. However, no cell subtype-specific EV markers have been identified in different neuronal and glial cell populations. Exosome-mediated communication may facilitate the anterograde and retrograde transfer of information across synapses. Exosomes that have been internalised are likely to alter post-transcriptional mRNA trafficking and translation, as well as induce local changes in synaptic plasticity.

Goldie et al. showed that a specific decrease in miRNA expression was observed in the neurites of potassium-depolarized cells, while exosomes generated by these cells were enriched in miRNAs and microtubule-associated protein 1B (MAP1B) [120]. Four of these miRNAs (miR-638, -149*, -4281, and let-7e) were observed to be negatively regulated by repetitive neuronal depolarisation. Interestingly, these miRNAs regulate the expression of mRNAs involved in synaptic plasticity. MAP1B is known to play an essential role in axon guidance, neuronal regeneration, and neurite branching. MAP1B also regulates the morphology of postsynaptic spines on the dendrites of glutamatergic neurons [120,121]. In addition, the nucleic acid content of neuron-derived EVs (nEVs) appears to be involved in several processes in CNS-like neurological development. For example, miR-132, a common neuronal EV miRNA that targets Ctbp2 (C-terminal binding protein 2), which is involved in regulating the Notch signalling cascade, leads to the accumulation of glial progenitor cells [122]. Neuron-derived extracellular vesicles appear to be involved in retrograde signalling by transferring synaptotagmin4 (Syt4). Syt4 is a membrane-trafficking protein whose expression is regulated by neuronal activity. Stg4 activation induces a signalling cascade involving cAMP, resulting in presynaptic stimulation. Syt4 is found in brain-derived neurotrophic factor-containing vesicles in hippocampal neurons, where it regulates synaptic plasticity and memory formation.

In addition, the postsynaptic release of Syt4-containing exosomes may regulate presynaptic quantal release of neurotransmitters, thereby facilitating synaptic tuning [123,124].

The presence of AMPA receptors in neuron-derived EVs strongly suggests that they have a role in modulating synaptic plasticity. As exosomes can fuse with the cell membrane of postsynaptic neurons, adding functional AMPA receptors to the postsynaptic button will further modulate synaptic strength. The presence of glutamate receptor subunits in neuronal exosomes implies that other ion channels may also travel between neurons, influencing their intrinsic properties [125].

Neurons and astrocytes communicate with through a variety of mechanisms, including exosome-mediated miRNAs transfer. This can regulate protein expression in parasynaptic astrocytes, which in turn can modulate synaptic function and neurotransmission. For example, exosomes carrying small RNAs and miR-124a have been isolated from neuron-conditioned media. They have been shown to be internalised by primary astrocytes. This increases the expression of miR-124a and glutamate transporter 1 (GLT-1) protein in the targeted cells [126]. GLT-1 (also known as excitatory amino acid transporter 2 (EAAT-2)) is crucial for the homeostatic maintenance of synaptic glutamate levels and the prevention of neuronal excitotoxicity [127]. As a result, the cargo of neuronal exosomes may contain complementary combinations of proteins and miRNAs that help astrocytes to maintain the homeostasis of neurotransmission in the CNS [128]. Neuron–microglia communication also occurs via neuron-derived exosome secretion. EVs can stimulate synaptic pruning by upregulating complement factors in microglia [121]. Bahrini et al. [129] induced neuronal degeneration after stimulating neuronal differentiation and synapse formation in PC12 cells. When PC12 cells were co-cultured with a mouse microglial cell line (MG6), microglia engulfed and phagocytized PC12 neurites.

After depolarisation, the MG6 cells were also pre-incubated with exosomes derived from differentiated PC12 neurons, indicating an increased clearance of degenerating neurites. These findings suggest that neuron-derived exosomes increased the expression levels of complement component 3 mRNA expression in MG6 cells rather than the direct transfer of C3 mRNA from PC12 cells [129]. These findings highlight the role of exosomes in the regulation of synapse elimination and reveal a novel mechanism by which active synapses promote the pruning of inactive ones by stimulating microglial phagocytosis with exosomes.

## 5. Astrocyte-Derived EVs

Astrocytes are CNS cells that play an important role in supporting neurons structurally and functionally, astrocytes are able to maintain the integrity of the BBB. Astrocytes secrete signalling molecules such as TGF-β and glial-derived neurotrophic factor (GDNF), which help to strengthen tight junctions between endothelial cells in brain capillaries [130]. Another function of astrocytes is the regulation of differential blood flow to cerebral microenvironments by secreting substances like (prostaglandins, adenosine, and Nitric Oxide) that alter the relative vasodilation–vasoconstriction balance in various neural networks [130]. In this section, we outline some of the important roles of astrocytes and the EVs they release. The various properties of these EVs, known as astrocyte-derived extracellular vesicles (ADEVs), and their potential role in brain health and neurological disorders are highlighted, with a special emphasis on emerging concepts. Astrocytes are one of the most abundant glial cell types in the CNS required for regulating brain function, development, homoeostasis, and defence [130]. They have been shown to provide neurotrophic support, be implicated in neurogenesis, synaptogenesis, and synaptic plasticity, and regulate the neurotransmitter clearance [131]. They regulate ions’ balance, maintain the BBB’s integrity and control different neuronal processes including the extracellular homeostasis [132]. In addition, astrocytes are major resident immune reactive cells in the CNS, mediating neuroinflammatory responses to infection and brain injury and playing a critical role in immunomodulatory responses [133]. Upon stimulation astrocytes release a large variety of cytokines, chemokines, reactive oxygen, and nitrogen species [134]. In fact, various cytokines, including TNF-α, IL-1β, and IL-6, as well as neurotrophic factors such as neurotrophic growth factor (NGF), BDNF, vascular endothelial growth factor (VEGF), and LIF (Leukaemia Inhibitory Factor) are secreted by stimulated astrocytes. Furthermore, upon astrocyte stimulation, chemokines such as CCL2, CCL20, and CXCL10, as well as antimicrobial peptides called β-defensins, were found to be upregulated. Reactive astrocytes express cell adhesion molecules (ICAM-1, VCAM-1), iNOS, and TLR3, with a neuroprotective response triggered by TLR3 activation, leading to the secretion of growth and differentiation mediators, along with pro- and anti-inflammatory cytokines. The type of stimulant present in the astrocytes’ microenvironment determines the content of these vesicles [135].

For example, astrocytes stimulated with lipopolysaccharide released bone morphogenetic protein 2 (BMP2), that promoted the differentiation of neural stem cells (NSCs) into astrocytes and inhibited axon remyelination in SCI lesions, and also released EVs enriched with miRNA-22-3p from NSCs [136].

Astrocytes stimulated with acidic fibroblast growth factor (aFGF) produced EVs which induce neuroprotection in AD pathology by enhancing neurite growth and reduction of Aβ loading on neurons in vitro [137].

Human primary astrocytes stimulated with morphine release EVs enriched with miR-138 that, in turn, activate microglia [138].

A recent study has shown that astrocyte-derived microparticles, also known as microvesicles, belonging to the spectrum of EVs were shown to initiate the NF-κB-mediated neuroinflammatory cycle following an acute exposure to CO. The astrocyte-derived MPs expressing TSP1 were able to establish the feed-forward neuroinflammatory cycle involving an interaction between CD36-NF-κB. All MPs-mediated events are blocked by the inhibition of NF-κB, myeloperoxidase, and anti-TSP-1 antibodies, indicating possible therapeutic targets for CO-induced neurological sequelae [139]. Beyond their supportive roles for neurons, astrocytes have been shown to play an important role in modulating neurodegeneration associated with different conditions such as Huntington’s disease (HD), Parkinson’s disease (PD), Alzheimer’s disease (AD), and amyotrophic lateral sclerosis (ALS) [140,141,142]. Due to the localized nature of these disorders within specific neural cells and CNS regions, identifying appropriate biomarkers for neurodegenerative disorders remains challenging. This suggests that exploring the molecular characteristics of EVs released by astrocytes is warranted for potential biomarker discovery in neurodegenerative disorders and as a promising vector for transferring biological messages between various nerve cells and neurons in the CNS. Recent evidence from the literature has highlighted the importance of the interaction between ADEVs and the different neurons and nerve cells in various physiological and pathophysiological conditions, since ADEVs can influence the function of recipient cells, including neurons, with both neuroprotective and detrimental effects to health and disease [143].

Recent evidence suggests the involvement of astrocytes in maintaining brain homeostasis and particularly in response to inflammatory conditions. However, the precise mechanisms of trans-cellular communication by astrocytes remains unclear. A recent study focused on characterizing the cargo proteins of ADEVs derived from human primary astrocytes under physiological and inflammatory conditions. In this study, the investigation of the proteomic profiling reveals a significant up-regulation of proteins in IL-1β-stimulated ADEVs compared to controls, indicating their involvement in cellular processes such as metabolism, organization, communication, and inflammatory response. The fluorescently labelled ADEVs administered into cultured mouse cortical neurons demonstrates a notably increased uptake of IL-1β-stimulated ADEVs compared to control ADEVs. This heightened neuronal uptake is attributed to the enrichment of surface proteins in IL-1β-stimulated ADEVs, confirmed by partial suppression with a specific integrin inhibitor. Furthermore, the treatment of neurons with IL-1β-stimulated ADEVs results in reduced neurite outgrowth, branching, and neuronal firing. These findings provide insights into the molecular mechanisms underlying the effects of ADEVs on neural processes, particularly in the context of inflammatory conditions [131]. ADEVs transfer proteins, nucleic acids, lipids, and other molecules between various cell types, influencing synaptic plasticity, neuron–glia interactions, neuroprotection, neurodegeneration, and the transfer of neuropathological molecules. When compared to naïve astrocytes, stimulated astrocytes release EVs carrying neuroprotective molecules such as synapsin 1, heat shock proteins, miRNAs, and glutamate transporters. It is worth highlighting that ADEVs generated in pre-determined conditions show therapeutic properties against various conditions like brain conditions and certain neurodegenerative diseases. ADEVs may help in the identification of the severity of neurodegeneration and providing a route to measure the healing stages in non-degeneration. However, these ADEVs may also exacerbate or mediate pathological processes, such as: PD, ALS, and other neuroinflammatory disorders [133]. For example, ADEV may dysregulate complement system in AD, stimulate leukocytes migration into the brain in inflammatory settings, and mediate the toxicity of motoneurons in ALS. The investigation of approaches involving the characterization of ADEVs through the activation of astrocytes or modulating their activities would have a direct effect on the ADEVs’ compositions and the messages they transfer to the target cells. It was also suggested that the regular analysis of the circulating ADEVs in the patients’ blood would be beneficial in the early detection of astrocyte-specific biomarkers and monitoring the neurodegenerative disorders progression. Additionally, it was recently reported that ADEVs were implicated in age-related changes in CNS function [144].

A recent study by Zhao et al. has reported that ADEVs provide neuroprotection in ischemic stroke patients by controlling autophagy and the release of miR-92b-3p, alleviating oxygen-glucose deprivation-induced neuron apoptosis, and inhibiting the expression of TNF-α, IL-6, and IL-1, resulting in a reduction in infarct volumes, highlight the potential of ADEVs as both diagnostic biomarkers and therapeutic targets for neurological disorders [145,146].

Stress is not only associated with a wide range of neuropsychiatric disorders, such as major depressive disorders, anxiety, schizophrenia, and post-traumatic stress disorder, but also can alter the onset and the progress of certain neurological disorders including neurodegenerative diseases [147]. Recent investigation has reported that patients with stress-induced exhaustion disorder had a significantly higher concentration of ADEVs when compared to healthy control patients with major depressive disorders. As a result, the release, and the leakage of ADEVs through the BBB is similar to what has previously been shown in patients following traumatic brain injury (TBI) [148,149,150].

Recent study by Zhang et al. investigated the neuroprotective effects of ADEVs and the subjacent mechanisms in a rat and mouse models of TBI called controlled cortical impact injury (CCI) [151].

In this study, ADEVs were isolated from primary astrocyte cultures from the cerebral cortices of new-born Sprague Dawley rats, demonstrating the presence of EV-specific markers (CD9, CD63, and CD81). ADEVs were injected 30 min post-controlled cortical impact (CCI) via the rat’s tail vein. The TBI severity was assessed using a modified neurological severity score (mNSS) at various time points post-TBI, and revealed lower mNSS and improved forelimb function in rats treated with ADEVs compared to other TBI groups. The attenuated neuronal injury involved the suppression of mitochondrial oxidative stress and apoptosis via the Nrf2 pathway. Additionally, ADEV-treated CCI rats exhibited an improved motor coordination in a rotarod test and better cognitive function in a water maze test. This suggests that early ADEV administration after CCI has significantly ameliorated both motor and cognitive functions following TBI [150,151].

On another hand, Varcianna et al. 2019 have investigated how the toxic phenotype of astrocytes contributes to motor neuron (MN) degeneration in ALS. In this study, induced astrocytes from ALS patients with C9orf72 mutations and non-affected donors were used to explore the impact of ADEVs on mouse motor neurons [77].

Intriguingly, these findings indicate dysregulation in ADEV formation and miRNA cargo in ALS in astrocytes, which consequently affected neurite network maintenance and MN survival in vitro. In particular, a downregulation of miR-494-3p levels, a negative regulator of semaphorin 3A (SEMA3A) and other axonal maintenance targets, was identified. Moreover, restoring miR-494-3p levels increased MN survival in vitro. Lower miR-494-3p levels were also observed in cortico-spinal tract tissues isolated from sporadic ALS donors. This highlights the potential therapeutic significance of this pathway in MNs. Overall, this suggests that ADEVs and their miRNA cargo play a role in MN death in ALS, where miR-494-3p is identified as a potential therapeutic target [152]. Further studies are required for investigating the emerging role of ADEVs in neuroprotection and its therapeutic potential, which appears to be critical for scientific breakthroughs in a range of diseases, including neurodegenerative diseases.

## 6. Microglia-Derived EVs

Microglia are resident immune cells in the brain, involved in both physiological functions and in pathological conditions [153,154]. Recent research on mice has reported that these cells originated from progenitors in the embryonic yolk sac early during development, which enter the CNS through the blood vasculature before the closure of the BBB [155]. Microglia cells are important for refining neural circuits in brain development, regulating synaptic networks, and promoting synaptic formation and maturation through synaptic pruning, neuronal apoptosis, neurogenesis, and the secretion of growth factors [156,157].

As macrophages, microglia can be classified as either classical (M1) or alternative (M2) and can switch between phenotypes. Infections or injuries can activate microglial cells towards the pro-inflammatory phenotype (M1), which produces pro-inflammatory mediators and induces inflammation and neurotoxicity. The alternative-anti-inflammatory-phenotype (M2) microglial cells release anti-inflammatory mediators and support neuroprotection. Different studies have demonstrated that balancing microglia M1/M2 polarization had promising therapeutic prospects in neurodegenerative diseases [158].

Our previous studies have reported that modulating pro-inflammatory polarization and consenting the shift towards anti-inflammatory polarization improved neuroinflammatory pictures in vivo [158,159,160].

In both physiological and pathological conditions, microglia communicate with neighbouring cells simultaneously through cell-to-cell contact, through the release of soluble factors, and via the exchange of biomolecules through secreted vesicles. It was demonstrated that microglial-derived EVs regulate synaptic transmission by inducing neuronal production of ceramide and sphingosine [41]. The enhancement in sphingolipid metabolism increases excitatory neurotransmission in vitro and in vivo, supporting the physiological modulation of synaptic activity by microglia [41]. Gabrielli et al. have found that one mechanism used by microglia-derived EVs to modulate presynaptic transmission is the endocannabinoid system [161].

Extracellular ATP is one of the most important stimuli for microglia vesicles shedding. When ATP is used to induce microglial EVs, these cells are enriched in IL-1β and GAPDH, which helps to regulate and propagate the neuroinflammatory response in the brain [162,163]. Since these vesicles were found to be loaded with functional P2X7 receptors, microglia may be able to reduce the amount of these receptors on the plasma membrane by shedding these vesicles, which would reduce P2X7-mediated apoptosis [164].

Additionally, in primary microglia cultures, as well as the BV2 cell line, serotonin binding to specific serotonin receptors (5-HT2a, b and 5-HT4) may promote the release of exosomes through an increase in cytosolic Ca^2+^ [165]. The recombinant Wnt3 also induces exosomes secretion through a GSK3-independent mechanism [166]. Microglia-derived EVs are also involved in neurodegenerative processes. For example, treating the BV2 cell line with alpha-synuclein increases the secretion of EVs, which express higher surface levels of MHC class II molecules and TNF-α, and have a higher capacity to induce neuronal apoptosis. Alpha-synucleinis is the small (14 KDa) acidic protein expressed in neurons of the central and peripheral nervous system as well as in blood cells and other tissues, it is a key component of Lewy bodies, in which it is found in an aggregated and fibrillar form [167]. LPS stimulation causes microvesicles rich in the proinflammatory cytokine IL-1β and the microRNA miR-155 to be released in microglia; these extracellular vesicles then intensify the proinflammatory reactions [168,169].

As shown in our previous in vitro study, EVs of BV2 cells produced with LPS have the ability to activate microglia in a way that is comparable to LPS polarizing microglia towards a pro-inflammatory state [170].

Other stimuli that induce EV secretion from cultured microglia are: pro-inflammatory cytokines [171,172], IL-4 [173,174], capsaicin [175,176], ethanol [177], and manganese [178].

Therefore, microglia EVs play a key role in cell–cell communication, altering the phenotype of recipient cells, and they may be vehicles for agents that propagate neuroinflammation in the CNS [179].

## 7. Oligodendrocytes and EVs

Glial cells known as Oligodendrocytes (OLs), are one of the main cell types in the white matter of the brain. They are considered as a key regulator of neuronal function in CNS development, homeostasis, and regeneration. They improve white matter function by generating myelin sheaths. The primary function of oligodendrocytes is to make myelin, which is a complex, organized, and tightly arranged myelin sheath wrapped around the CNS axons to favour impulse conduction [180]. Moreover, the axonal myelin unit contributes to metabolic and nutritional supply, which helps neurons to maintain their function [180]. According to Greenhalgh et al., oligodendrocytes may have a potential immunomodulatory ability [181]. Oligodendrocytes need to be supported by adjacent cells such as neurons [182], astrocytes [183], microglia [184] and cerebrovascular endothelial cells [185] to play a role in the white matter.

Oligodendrocytes incorporate both internal and external stimuli to adapt myelination patterns, and thus receiving these signals requires precise control composition. In the absence of neurons, primary cultured oligodendrocytes release after Ca^2+^ treatment EVs that transport unprocessed proteolipid protein (PLP) and DM20, two abundant myelin proteins which are generated by alternative splicing from the PLP gene [2].

Oligodendroglial EVs may provide trophic support to axons and may even help with myelin clearance. EVs isolated from the rat primary oligodendrocytes demonstrated an autoregulatory effect on recipient oligodendrocytes, resulting in a decrease in cell surface expansion and myelination via Rho-ROCK-myosin interactions. Interestingly, Interestingly, the pro-myelinating impact of neuronal conditioned media, which contains neuronal components that govern myelination, was reversed by these oligodendrocytes-derived EVs [186].

Another role of oligodendrocyte-derived EVs is neuronal protection; in fact, they improve neuronal survival under conditions of oxygen-glucose deprivation (OGD), an in vitro model of stroke. The oligodendrocyte-derived EVs can activate prosurvival signalling pathways in neurons such as AKT, Erk1/2, and JNK. Moreover, they are able to transfer stress-protective enzymes such as superoxide dismutase 1 (SOD1), Hsc/Hsp70 protein, and catalase [186,187,188].

The neuron protection mediated by oligodendrocyte-derived EVs is due also to an increased phosphorylation of the transcription factor CREB and the enzymes GSK-3α/β and gSK3β [186].

These EVs did not promote neuron regeneration and did not provide neuroprotection when administered after the injury [2,186,187].

## 8. Future Directions

In this review we have highlighted the new emerging concepts and provided a recent update about EVs with an emphasis on their enormous potential in neuron–glia communication.

Through transferring their cargo, EVs carry out a wide range of homeostatic functions in the central nervous system (CNS), such as regulating neuronal excitability, synaptic plasticity, myelination, microglial activation, and neuronal and glial response to stimuli (Figure 2) (Table 3). Future research will predominantly focus on cell engineering and manipulating cells in laboratories to stimulate the generation of distinct EVs with therapeutic proprieties, especially in the field of CNS diseases. In particular, additional studies will help to understand the mechanism leading to EVs formation in CNS, the EV cargo in relation to stimuli, and also the presence proteins localized to extracellular membrane that may be useful in mediating the recognition of target cells.

## Figures and Tables

**Figure 1 ijms-25-01629-f001:**
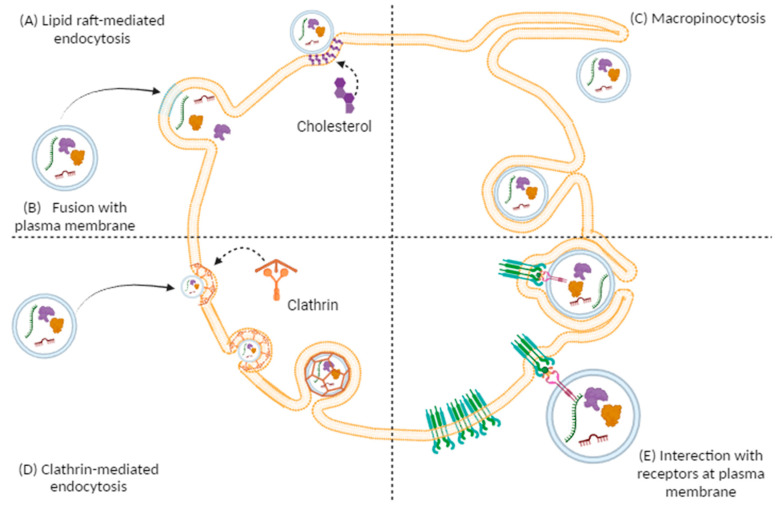
Schematic of the different mechanisms utilized to internalise extracellular vesicles into recipient cells: (**A**) lipid raft-mediated endocytosis; (**B**) fusion at the plasma membrane; (**C**) phagocytosis; (**D**) clathrin-mediated endocytosis; and (**E**) interaction with receptors on the plasma membrane.

**Figure 2 ijms-25-01629-f002:**
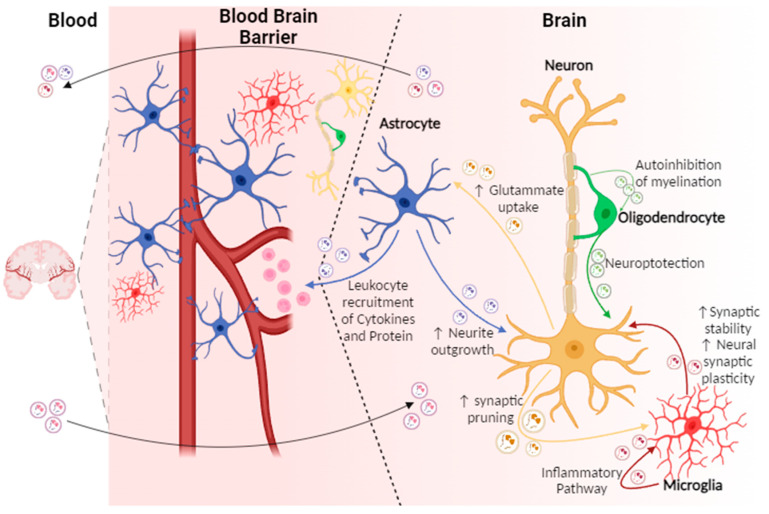
The figure illustrates the physiological role of extracellular vesicles in the central nervous system. EVs are able to transfer their contents between the different pairs of cell types shown in the figure: astrocytes, neurons, oligodendrocytes, and microglia. The arrows indicate the direction of transfer and are labelled with the content of EVs commonly found for this cell-to-cell communication pathway and some of the functional roles identified for the pathway. Microglial EVs can lead to increased synaptic stability and neuronal synaptic plasticity. Neurons increase glutamate uptake and synaptic pruning. astrocytes release EVs that induce neurite outgrowth. EVs are also derived from oligodendrocytes and contain RNA and protein cargoes that confer neuroprotection to neurons. EVs can also cross the blood–brain barrier (BBB) in both directions to enable communication with peripheral targets.

**Table 1 ijms-25-01629-t001:** Extracellular Vesicles miRNA in Central Nervous System.

EVs miRNA Name	EVs Origin	Target Pathways/CNS Component	Major Function	Ref.
miR-15a miR-15b	Endothelial Cells	Microglia	Induce Microgliosis Elevate Levels of Mir-155 Increase TNF-α And Il-6	[10,68]
miR-21	Endothelial Cells	SNC, Microglia	Induce Microgliosis Elevate Levels of Mir-155 Increase TNF-α and Il-6	[10,69]
miR-21-5p	Neuron	Microglia, Astrocytes, Endothelial Cells	Modulation of Microglial Activity Pro-Inflammatory Responses Gene Transcription in Astrocytes Brain Vascular Integrity	[70]
miR-92b-3p	Neuron	Neuron	Alleviating Oxygen and Glucose Deprivation-Induced Neuron Apoptosis Inhibiting the Expression Of TNF-α, Il-6, Il-1 Reduction in Infarct Volumes	[71]
miR-124	Neuron	Microglia, Astrocytes, Endothelial Cells	Modulation of Microglial Activity Pro-Inflammatory Responses Gene Transcription in Astrocytes Brain Vascular Integrity	[72]
miR-124a	Neuron	Astrocytes	Increases GLT-1 Homeostatic Maintenance of Synaptic Glutamate Levels Prevention of Neuronal Excitotoxicity	[73]
miR-132	Neuron	Ctbp2 on glial progenitor cells	Regulating the Notch Signalling Cascade	[74]
miR-155	Microglia	Microglia	Intensify the Proinflammatory Reactions Microgliosis	[75]
miR-494-3p	Astrocytes	Neuron	Negative Regulator of Semaphorin 3A	[76]

**Table 2 ijms-25-01629-t002:** The potential applications of extracellular vesicles in clinical trials related to neurological disorders.

EVs Type/Origin	Potential Application	Cargo/Delivered Drug	Study Identifier of Clinical Trials	Ref.
MSCs	Alzheimer disease	No	NCT04388982	[91]
MSCs	Stroke/Cerebrovascular disease	miR-124	NCT03384433	[92]
Multiple origins	Stroke	Prognostic biomarkers, profiling biomarkers	NCT05370105	Ongoing
Neurons, astrocytes, microglia and oligodendrocytes	Neurodegenerative diseases (Alzheimer’s disease)	Circulating biomarkers	NA	[93]
Multiple origins (blood)	Neurological disorders and other disorders	Circulating biomarkers	NA	[94]
Multiple origins (saliva)	Parkinson disease	Saliva-based biomarker	NCT05320250	Ongoing
Multiple origins (blood)	brain Huntingtin (HTT)	Blood-based biomarker of disease progression or conversion	NCT06082713	Ongoing
Multiple origins (blood)	meningioma	Biomarkers of (early) tumour progression. DNA methylation profiling of plasma EVs	NCT06104930	Ongoing
Multiple origins (blood)	Traumatic brain injury	Early disease assessment and biomarker of prognosis following traumatic brain injury;HMGB1	NCT05279599	[95,96,97]
Multiple origins (blood, CSF)	Alzheimer’s disease, neurofibrillary degeneration (NFD)	Detection of early markers of (NFD) such as Tau, therapeutic targets	NCT03381482	Ongoing

NA: not applicable.

**Table 3 ijms-25-01629-t003:** Effects of extracellular vesicles from central nervous system cells.

Origin	Effects	Ref.
Neuron-derived EVs	Synapse maintenance in central neurons.	[110]
	Synaptogenesis, synapse, and dendrite maintenance, spatial learning and memory, formation of hippocampal long-term potentiation.	[167,168]
	Axon guidance and synapse formation.	[113]
	Neurite outgrowth, neuronal survival, and death.	[114]
	Modulation of microglial activity	[116,117,119,120,121,126]
	Synaptic plasticity, helping astrocytes to maintain the homeostasis of neurotransmission.	[122,128]
	Formation of excitatory synapse in recipient neurons	[118]
	Synaptic plasticity, memory formation, facilitating synaptic tuning.	[111,123,124]
	Prevention of neuronal excitotoxicity	[127]
Astrocyte-derived EVs	Helping to strengthen tight junctions between endothelial cells in brain capillaries.	[123,131]
	Neurogenesis, synaptogenesis, and synaptic plasticity.	[135]
	Mediation of pathological processes and neuroinflammatory disorders.	[145,146]
	Neuroprotection.	[133,152]
Microglia-derived EVs	Regulation and propagation of neuroinflammatory response	[156,157,158,162,163]
	Modulation of inflammatory mediators	[156,157,158,162,163]
	Support neuroprotection	[41]
Oligodendrocytes	Helping in myelin clereance	[185]
	Transfer stress-protective enzymes	[186,187,188]
	Neuron protection	[186]

## Data Availability

Not applicable.

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
