# Peer review of "Extracellular Vesicles in the Central Nervous System: A Novel Mechanism of Neuronal Cell Communication"

_ijms, 2024, doi:10.3390/ijms25031629_

Round 1

Reviewer 1 Report

Comments and Suggestions for Authors

The authors presented an overview of extracellular vesicles role in neuronal cell communication in the central nervous system. Review paper is comprehensive and summarizes novel research from extracellular vesicles and neurological field of science. Listed below are recommendations for authors to improve the clarity of the manuscript.

1.       Line 26. Please revise the sentence.

2.       Line 123. Verb missing.

3.       Line 131. Add Figure 1. at a beginning of the figure legend. Same applies to Figure 2 legend.

4.       Line 142. Please provide a full name for ESCRT when it is first mentioned in the manuscript.

5.       Line 149. Please add neutral before sphingomyelinase (nSmase).

6.       Line 150. The nSMase and ___?___ forms membrane microdomains that facilitate inward budding [34].

7.       Line 174. Please revise this sentence. Also, it seems more suitable if sentence (line 177) is mentioned before Line 174.  It seems like authors jumped to the new topic so a brief introduction of the reader to Wnt protein and signalling could be of great value.

8.       The same observation as before. Line 183. seems more suitable if added before Line 180. If not please add the full name for PRR7 when first mentioned in text.

9.       Line 186. “Are” AMPA receptors.

10.   Line 210. Please revise the sentence.

11.   Line 221. Please revise the sentence.

12.   Line 222. Please revise the sentence.

13.   Line 235. “a specific” has been written twice.

14.   Line 287. Word important is misspelled.

15.   Line 315. If available in literature, could authors elaborate in more detail which specific stimulus induces release of different vesicle content from astrocytes (and what kind of content).

16.    Line 384. Please check if the statement is correct. “Healthy control of patients with major depressive disorders”

17.   Line 414. Potential is mentioned twice.

18.   Line 443. By instead of my.

19.   Line 453. “release” is mentioned twice and add + to Ca2+

20.   Line 456. Additional information on alpha-synuclein would be more informative for the reader.

21.   Line 488 and 494. Statements are repetitive. Please revise.

22.   In section 7. Future directions (please revise the numbering in the manuscript), add references if applicable.

Comments on the Quality of English Language

Minor editing of English language required. Check for missing or excess of spaces, misspelling and grammatical errors.  

Author Response

We thank the reviewer, which gave us the opportunity, with its suggestions, to improve the quality of the manuscript and add further useful information on the field. We have carefully taken their comments into consideration in preparing our revision, resulting, as we hope, in a paper clearer, more compelling, and broader.

In the attached file the answers.

Reviewer 2 Report

Comments and Suggestions for Authors

In the present review, Filannino et al. described the role of EVs in cell-cell communication in the CNS. The review is well-written and full of useful information to the readers in the field. However, a review article is not about providing the information only. The following areas are missing in the present version of the manuscript and should be taken care of. My comments are as follows:

Major:

1. The biogenetic mechanism of different forms of EVs should be discussed.

2. Page 4, Line 149: Other than nSMase, acid sphingomyelinase is also known for releasing the EVs. It should be mentioned along with the following citation: PMID: 37875382.

3. It would be nice to include a table describing neuronal, astrocyte, microglia, and oligodendrocytes EVs along with the references.

4. Briefly, discuss the role of EVs in developmental pathology of CNS.

5. Mention the therapeutic potential of EVs in CNS pathology.

6. Briefly mention how EVs could play a role as CNS biomarker.

7. Summarize the potential of EVs-based therapies against CNS pathology.

8. Briefly mention EVs application in clinical trials in a tabular form.

Minor comments:

1. A few grammatical errors remain which needs to be corrected.

2. Abstract: line 23: EVs also contain metabolites... please include.

3. Abstract: line 25-26: remove the full form of EVs as it already been mentioned before.

4. Introduction: line 42: it should be "to circulate" rather than "circulate".

5. Introduction: line 40-41: it should be "broadly subdivided" instead of "subdivided" as other forms of the EVs have recently come up.   

Comments on the Quality of English Language

Minor editing is required.

Author Response

(The authors gave the same response as above.)

Reviewer 3 Report

Comments and Suggestions for Authors

This current article entitled “Extracellular Vesicles in the Central Nervous System: A Novel Mechanism of Neuronal Cell Communication” by Filannino et al was devoted to analyzing the role that extracellular vesicles (EVs) play in CNS cell communication, with particular emphasis on how EVs from different CNS cell types to maintaining or altering the CNS homeostasis. The article is so interesting and highly demanding in current research. The manuscript can be accepted only after addressing the following queries:

1.      To attract broader readers, the authors should provide a graphical abstract by proposing how exosomes cross the BBB and release the cargoes in CNS.

2.      In Figure 1, where the authors discuss extracellular vesicles to be internalized into recipient cells, there is inadequate discussion about this topic in the review. This raises questions about why this figure was included in the article.

 3.      Add references for Line no.- 46-47.

 4.      Line no.-43. Authors may include a recently published article: ACS Chemical Neuroscience 14 (17), 2981-2994.

5.      Referencing missing for the Line no.-49-51.

6.      Add some more excellent references (for Line- recently published elsewhere: ACS Biomaterials Science & Engineering 9 (2), 577–594, IJS Global Health 6 (3), e0141)

 7.      Authors can include the following reference along with ref no- 25. ACS Biomater. Sci. Eng. 2023, 9, 9, 5205–5221.

 8.      Several text formatting issues and typo mistakes were observed throughout the manuscript, e.g. unwanted paragraphs, started after writing one/two sentences, space missing between two numbers [38,39 & 41,42 and so on].

 9.      Fig. 1: Expected more clear and high DPI image; especially text in the figure.

 10.   Provide a tabulation form of the extracellular vesicles-derived miRNAs, effector factors, and their role in CNS.

 11.   The review does not discuss other forms of nucleic acid transfer, such as lncRNA or circular RNA, from exosomes that contribute to cellular communication in CNS. This is an important aspect that should be addressed in the review.

 12.   The conclusion is to be concise and address the lack of current research, and proper future direction.

Comments on the Quality of English Language

Some sentences need to be revised. Several text formatting issues and typo mistakes were observed throughout the manuscript, e.g. unwanted paragraphs, started after writing one/two sentences, space missing between two numbers [38,39 & 41,42 and so on].

Author Response

(The authors gave the same response as above.)

Round 2

Reviewer 2 Report

Comments and Suggestions for Authors

The authors adequately addressed all my concerns. I have no further comments.

Reviewer 3 Report

Comments and Suggestions for Authors

This current article entitled “Extracellular Vesicles in the Central Nervous System: A Novel Mechanism of Neuronal Cell Communication” by Filannino et al was devoted to analyzing the role that extracellular vesicles (EVs) play in CNS cell communication, with particular emphasis on how EVs from different CNS cell types to maintaining or altering the CNS homeostasis. The article is so interesting and highly demanding in current research. The revised manuscript improved a lot and addressed all the queries. I recommend to accepting this article in its current form.